# The Effect of Tartary Buckwheat Flavonoids in Inhibiting the Proliferation of MGC80-3 Cells during Seed Germination

**DOI:** 10.3390/molecules24173092

**Published:** 2019-08-26

**Authors:** Xiao-Li Zhou, Zhi-Dong Chen, Yi-Ming Zhou, Rong-Hua Shi, Zong-Jie Li

**Affiliations:** School of Perfume and Aroma Technology, Shanghai Institute of Technology, Shanghai 201418, China

**Keywords:** tartary buckwheat, flavonoids, MGC80-3, flow cytometry, western blot

## Abstract

Tartary buckwheat (*Fagopyrum tataricum* (L.) Gaertn) is rich in functional compounds such as rutin, quercetin, d-chiro-inositol, dietary fiber, and essential amino acids. Electric field (EF) treatment before sprout germination results in physiological and chemical changes, and some alterations might lead to positive applications in plant seeds. MTT assay showed that the effect of total flavonoids on human gastric cancer cell line MGC80-3 was significantly changed after EF treatment for different germination days (3–7 days). Among them, the total flavonoids of tartary buckwheat (BWTF) on the third day had the most obvious inhibitory effect on MGC80-3 (*p* < 0.01). In addition, flow cytometry evidenced that different ratios of quercetin and rutin had effects on the proliferation of MGC80-3. The same content of quercetin and rutin had the best effect, reaching 6.18 ± 0.82%. The anti-cancer mechanism was mainly promoted by promoting the expression of apoptotic proteins. The expression of Bax/Bcl-2 and caspase-8 in MGC80-3 cells was mediated by BWTFs. This study has good research value for improving the biological and economic value of tartary buckwheat.

## 1. Introduction

As a unique variety of buckwheat in China, tartary buckwheat (*Fagopyrum tataricum* (L.) Gaertn.) has received increasing attention because of its excellent properties, such as anti-oxidation, inhibition of tumors, etc. These properties are related to its unique bioactive composition, which includes phenols, proteins, and so on. Interestingly, flavonoids such as rutin or quercetin have not been found in any cereals or pseudocereals except buckwheat [1]. In particular, there is a growing interest in tartary buckwheat sprouts as a healthy food source for these flavonoids, which are produced at high levels in sprouts due to germination [2]. Previous research has demonstrated that grain germination as a biochemical technique can effectively improve nutrient content and reduce antinutrients in cereals [3]. Therefore, germination has been widely used for its ability to decrease levels of antinutritional factors present in seeds, at the same time improving the concentration and bioavailability of their nutrients [4].

Flavonoid compounds belong to a family of plant secondary metabolites important in responses to both biotic and abiotic stresses. Flavonoid biosynthesis consists of multiple chemical reactions and pathways containing several enzymes. Of these, two key enzymes, phenylalanine ammonia lyase (PAL) and chalcone isomerase (CHI), play critical roles in catalysis [5]. PAL is the first rate-limited enzyme in the reaction pathway, which converts phenylalanine into cinnamic acid. Therefore, the amount of PAL and its intracellular enzyme activity determine flavonoid production [6]. CHI catalyzes the transformation reaction of naringenin chalcone into naringenin. The second-order rate constant for flavonoid synthesis by CHI indicates that the CHI-catalyzed reaction is diffusion-limited [7].

During the normal germination process, the activities of enzymes change in real time, which makes organisms develop and grow normally. However, with external stimuli such as electric fields, some enzymes activities may be changed, leading to variations in the various active substances in organisms. In recent years, reports have shown that levels of several nutrients, such as flavonoids, γ-aminobutyric acid, and some enzymes, increase to adapt to adverse environmental conditions, such as hypoxia, temperature stress, drought, or additive stress [8]. Imani [9] reported that the movement of electrons, ions, and other species was influenced by electric field (EF) treatment, which changes cellular metabolism and may positively affect the growth of the plant. These results demonstrate that pulsed electric field treatment of imbibed seeds can stimulate changes in metabolism in the resultant seedlings, increasing the bioprotective potential of their shoots/sprouts and hence value as functional foods [10].

A great deal of work has focused on the influence of flavonoids on the physiological functions of humans. In addition, flavanoids have anti-inflammatory, anti-viral, and anti-cancer effects. Some studies have shown that the anticancer effect of total flavonoids may due to the regulation of immune function and the inhibition of inflammatory cytokine production.

There is a high incidence of gastric cancer, and occurrence of cancer is a multi-process event which involves a range of biological activities, including genetic and epigenetic changes. At present, treatment of cancer is not very efficient, so the prevention of cancer is particularly important. Studies over the past years have found that the cells can be protected by flavonoids, which act against cancers caused by intracellular injury [11]. The inhibitory effect of flavonoids on cancer cell proliferation has also been proven in many studies. Veeriah et al. [12] described that the flavonoids extracted from apples can inhibit colon cancer by the expression of a differential gene. Zhang et al. [13] found the treatment with citrus flavonoid nobiletin resulted in an upregulation of the antiapoptotic protein and downregulation of the antiapoptotic proteins bcl-2 and p53, which lead to human pancreatic carcinoma cells. Elkady et al. [14] reported that the medicinal herb *Nigella sativa* flavonoid can inhibit proliferation and induced apoptosis in MCF-7 cells. In conclusion, flavonoids from plants probably reduce cancer risk.

In order to improve the economic value of tartary buckwheat, in this report, EF was used to enhance the flavonoid content in malted buckwheat seeds. The biological activity of the total flavonoids of tartary buckwheat treated by EF was verified using human gastric cancer cells MGC80-3. MTT, flow cytometry, and Western blot were used to test the mechanism of tartary buckwheat flavonoids inhibiting MGC80-3 cells. Tartary buckwheat has already been proved to high value for medicine and health. If enhancement effects of flavonoids content can be tested by novel germination methods, tartary buckwheat could be explored for functional food and clinical applications.

## 2. Results

### 2.1. Effect of Electric Field on the Germination of Tartary Buckwheat Seeds

The contents of total flavonoids in tartary buckwheat (BWTF) were measured in this study, and the intensity and time-dependent effects of electric field (EF) treatment on the changes of the BWTFs were recorded and compared. As shown in Figure 1A(a), at 4–6 days of germination, the 10.0 kV/m-treated BWTFs were significantly different from the others (*p* < 0.05). The highest content reached 70.04 ± 6.17 mg/g. The flavonoid content of the treated seeds was statistically higher than that of the control group (*p* < 0.05). Moreover, the levels of total flavonoid expression was affected by EF treatment time (Figure 1A(b)). The total flavonoid contents in EF treatment groups were statistically higher than in the control, and the EF group treated for 15 min displayed significant differences on the fifth day. For the two key enzymes (PAL, CHI) in tartary buckwheat seeds the activities also increased continuously during germination (1–7 days) and had the same trend as BWTFs after EF treatment (Figure 1B). Overall, the EF stimulation can affect enzyme activity and has a certain promoting effect on the production of total flavonoids. However, as the enzyme activity changes, the content of the total flavonoids may increase, and the composition may change accordingly. This may lead to different bioactivities of total flavonoids in tartary buckwheat at different germination stages.

### 2.2. Quantitative Analysis of BWTFs

Flavonoids are important secondary metabolites in higher plants, whereby which quercetin and rutin are unique in tartary buckwheat [15]. Therefore, rutin and quercetin were used as reference. The composition changes were measured by high-performance liquid chromatography (HPLC). The contents of rutin and quercetin in tartary buckwheat were detected by HPLC. The results are shown in Figure 2. At the beginning, the content of quercetin was higher than that of rutin. With the increase in germination days, the content of rutin increased gradually while the quercetin content decreased, and the trend of both became stable, whereby which the levels of rutin remained at 36.67 ± 1.44 mg/g and quercetin content remained at 11.16 ± 2.48 mg/g. On the third day of germination, the contents of quercetin and rutin were closely similar. Changes of the components in tartary buckwheat may be related to the enzyme activity during seed germination. This is similar to the change in the content of quercetin and rutin during normal germination, but the content of rutin and quercetin was significantly increased after the EF treatment of tartary buckwheat seeds. With the changes of composition in total flavonoids, the function of BWTFs may also change.

### 2.3. Anti-Tumor Activities of Buckwheat Flavonoids

The MGC80-3 cell line was used to study the proliferation inhibitory effects of total flavonoids in tartary buckwheat seeds during germination. Figure 3 shows the results of the MGC80-3 survival status. BWTFs extracted on different germinating dates were tested for anti-tumor activity. Among seven samples of tartary buckwheat total flavonoids (BWTFs), the total flavonoids of tartary buckwheat on the third day (BWTF3) exhibited the lowest cell viability (47.3 ± 6.8%, *p* < 0.05). BWTF4, BWTF6, and BWTF7 were second, whereas nonsignificant differences were observed between them (54.4 ± 8.7%, 53.2 ± 8.1%, and 50.3 ± 7.4%, *p* > 0.05, respectively) (Figure 3A). In order to further explore the inhibitory effect of BWTFS on MGC80-3, the 50% inhibiting concentration was determined. The results show that BWTF3 has significant differences compared to BWTF4, BWTF6, and BWTF7, which indicated the best inhibitory effect on MGC80-3 (14 ± 0.41 μg/mL, *p* < 0.05) (Figure 3B). The different effects of tartary buckwheat total flavonoids on MGC80-3 cell proliferation may be caused by different proportions of rutin and quercetin flavone molecules.

### 2.4. Effects of Quercetin and Rutin on the Proliferation of MGC80-3 Cells

The inhibitory effects of quercetin, rutin, and QR (quercetin:rutin = 1:1) on MGC80-3 were studied, in which quercetin and rutin were the two main flavonoids in tartary buckwheat. As shown in Figure 4, with the concentration of the sample increased, the cell viability gradually reduced, indicating that the inhibition of quercetin and rutin on cancer cells is related to the concentration. QR was significantly different from rutin (*p* < 0.05). The inhibitory effect of quercetin on cells was better than that of rutin, especially when the concentration was 12.5 μg/mL (*p* < 0.05). The differential cytotoxic effects of quercetin and rutin may be caused by the different rates of their metabolism in the cell. The results of QR showed that the quercetin and rutin might have some synergistic effect in the cellular metabolism [16].

### 2.5. Apoptotic Assay

Apoptosis is a way in which the death of cells is controlled by gene. The differential staining effect of Hoechst and propidium iodide (PI) can detect normal, apoptotic, and necrotic cells. The quercetin and rutin standard samples were mixed in proportion which references the changes of content during the germination of tartary buckwheat. The inhibitory effects of different components of flavonoids on MGC80-3 cells were studied. The results are shown in Figure 5. With a close ratio of quercetin and rutin, the apoptotic rate was increased to the maximum of 6.18 ± 0.82% (Figure 5A(d), *p* < 0.05), and at the same concentrations, the apoptotic rate of BWTF3 was slightly lower than that of the Figure 5B(e), which may be due to the low purity sample during the process of extraction. Dimethyl sulfoxide (DMSO) was used as negative control (apoptosis rate, 1.8 ± 0.02%, *p* < 0.05) and 5-fluorouracil (5-FU) was used as positive control (apoptosis rate, 4.84 ± 0.57% *p* < 0.05). With changes in content of quercetin and rutin, the apoptotic rate of human gastric cancer cells also changes, which seems to indicate that quercetin and rutin are the main components of tartary buckwheat flavonoids that inhibit cancer and have some synergistic effects.

### 2.6. Protein Assay and Western Blotting

As the report of previous research, the caspases are important in the cell apoptosis [17], and caspase 8 belongs to mammalian caspase family, which plays an important role in the early apoptosis with the damage of the mitochondrial [18]. The anti-apoptotic (Bcl-2) and pro-apoptotic (Bax) protein expression, and caspase-8 activation of the buckwheat flavonoid bioactive substances (BWTF3) were determined by Western blotting to determine the inhibitory effect of flavonoids effect of apoptosis. The expression of Bcl-2 at the protein level was decreased, the ratio of Bax/Bcl-2 increased, and the expression of caspase-8 at the protein level was increased after treatment with quercetin and BWTF3 for 12 h (Figure 6).

## 3. Discussion

After treatment with electric field (EF) (7.5–12.5 kV/m, 5–25 min), the content of the total flavonoids of tartary buckwheat (BWTFs) increased significantly during the germination (1–7 days). The most suitable condition for EF treatment is 10 kV/m and 15 min. The contents of total flavonoids in tartary buckwheat were 70.03 ± 3.4 mg/g. Kim et al. [19] found that the content of rutin in treated tartary buckwheat was 21.8 mg/g after six days of germination. The same level on the fourth day after electric field treatment was found in this study. This may be related to the increase in enzyme activity due to the EF stimulation. The production of flavonoids is closely related to the enzymes in the seeds. During the biosynthesis of buckwheat flavonoids, four key enzymes play the critical role in the catalyzing process [5]. They are phenylalanine ammonia lyase (PAL), chalcone synthase (CHS), chalcone isomerase (CHI), and flavone synthase (FLS). PAL is the first rate-limited enzyme in the reaction pathway [20]. The expression amount of PAL and its intracellular enzyme activity determine the production of rutin and quercetin. CHI also plays an important role in the flavonoid pathway [21]. The biological effects of extra added EF on plant organs, tissues, and cells might be associated with electro-behavior of ion transporters, ion channels, and metal ion in the enzyme-active center. This electrochemical action might be related to the regularization of cell membrane potential and dipole moments of the enzyme-active site [22]. Although these biologic electric fields are extremely weak, their electrochemistry balance is importance for the normal metabolism of plant itself [23]. However, the specific mechanism still needs to be certified. The eventual stabilization may be due to competition with other enzymes, such as alpha-amylase [24].

With the increase of the total flavonoid content, the compositions are also changing, as shown in Figure 2. Quercetin gradually decreased and the trend of rutin shows the opposite. A similar content of quercetin and rutin was achieved on the third day of germination. The biological activity of BWTFs was affected by the changes in components (Figure 3). BWTFs can inhibit the proliferation of human gastric cancer cell line MGC80-3, and its inhibitory ability also shows differences because of the changes in composition. With further validation, the inhibitory effect of quercetin on MGC80-3 is better than rutin, and the quercetin and rutin complex has a better inhibitory effect on MGC80-3 cells. Similar conclusions were also shown by Suzuki et al. [2], that is, the anti-cancer effect of natural flavonoids was superior to that of the pure products.

Gastric carcinoma is a common malignancy. Flavonoids in plants are effective anticancer substances. Babu et al. [25] demonstrated that flavonoid from the *Syzygium alternifolium* significantly inhibited the cell proliferation of human gastric cancer cells. According to Figure 3, the total flavonoids isolated from the tartary buckwheat can inhibit the proliferation of MGC80-3 in vitro. The inhibitory effect of BWTF3 is significant. In recent years, many anticancer perspectives of flavonoids and mechanisms have been proposed, for example that they prompt p53-subordinate apoptosis, activate caspase-3 and caspase-9, down-regulate Bcl-2 and up-regulate Bax and cytochrome c, and that they cause mitochondrial apoptosis [26]. The inhibition of apoptotic programs may be attributed to Bcl-2 protein. The relative amounts of Bcl-2 and Bax proteins determine cell survival or death following an apoptotic stimulus. Caspase-8 is a key promoter in the death receptor-mediated apoptotic pathway and apoptosis relies on a cascade of caspases that are specific for aspartate [27]. Caspase-8 is able to self-cleave and activate by oligomerization, which activates the downstream cysteine protease and subsequently produces the apoptotic effect [28]. Thus, the effects of flavonoids (BWTF3) on the expression of Bcl-2, Bax, and caspase-8 at protein levels in MGC80-3 cells were detected to explore whether tartary buckwheat was involved in the process of cell apoptosis. The results of western blot revealed a significant alteration of the expression of Bax/Bcl-2 and caspase-8 at the protein level in MGC80-3 cells treated with BWTF3, suggesting that BWTF3 is involved in the apoptosis of MGC cells at the gene level (Figure 6).

## 4. Materials and Methods

### 4.1. Chemical and Reagents

Quercetin, rutin, dimethyl sulfoxide (DMSO) and 5-fluorouracil were purchased from Shanghai Kaiyang Biotechnology Company (Shanghai, China). Acetonitrile (HPLC), RPMI Medium 1640, and fetal bovine serum (FBS) were products of Gibco Life Technologies (NY, USA). The Apoptosis and Necrosis Assay Kit was purchased from the Beyotime; OLYMPUS IX73. Guava easyGyte; 3-(4,5-dimethylthiazol-2-yl)-2,5-diphenyltetrazoliumbromide (MTT), Cell lysis buffer for Western and IP, the SDS-PAGE Gel Quick Preparation Kit, Protein Marker, Polyvinylidene Fluoride (PVDF) Membrane, Blocking Buffer, Primary Antibody Dilution Buffer, Secondary Antibody Dilution Buffer, and BeyoECL Plus were purchased from the Beyotime Institute of Biotechnology (Shanghai, China).

### 4.2. Electrical Treatment and Germination of Buckwheat Seeds

Tartary buckwheat (*Fagopyrum tataricum* (L.) Gaertn) was cultivated in Shanxi province of China, the seeds were collected and treated by EF using different treating intensities and times. The germination of tartary buckwheat occurred in Shanghai. The germination methods of EF-treated buckwheat seeds are detailed elsewhere [29].

### 4.3. Extraction of Total Flavonoids Extracts from Germinated Buckwheat Seeds

Total flavonoids were extracted from germinated buckwheat seeds which treated by EF. The malted sprout powder was dissolved in 70% ethanol and extracted in a 70 °C water bath for 6 h [30]. The extracted solution was filtered, lyophilized and stored at −20 °C until further analysis.

### 4.4. Enzyme Activity Assay

The PAL and CHI assay was carried out as previous reported with minor modifications [31]. Generally, the reaction system was composed of 0.875 mL borate buffer and 0.25 mL crude enzyme solution, and then the assay was initiated by L-phenylalanine with 30 min incubation at 30 °C. Here, 50 mM boric acid were used as the control. The absorptance 290 nm was detected to examine the product of cinnamic acid. CHI activity was measured as described by McCallum [32]. Here, 1 mL crude enzyme solution (containing 7.5 mg/mL BSA and 50 mmol/L KCN) was mixed with 5 μL tetrahydroxy chalcone in a 2-ethoxyethanol water bath (34 °C, 5 min) enzyme solution for 10 min as the control. During the isomerisation process, the absorbance at 381 nmwas measured to calculate CHI activity.

### 4.5. Cell Culture

The human gastric cancer MGC80-3 cells were purchased from cell bank in Shanghai Institutes of Biological Sciences (SIBS) of Chinese Academy of Sciences (CAS). The method of cell culture was in accordance with previous studies, and the complete cell culture medium was RPMI 1640 with 10% fetal bovine serum (FBS) and 1% penicillin–streptomycin [33].

### 4.6. Cell Proliferation Inhibitory Activity

The effect of BWTFs on the viability of MGC80-3 was measured by MTT [34]. Human gastric cancer cells (1 × 10^4^ cells/100 μL) were transferred to 96-well plates and incubated for 24 h. Then the test was carried out after 24 h of intervention with BWTFs (50 μg/mL, 200 μL/well). MTT solution (10 μL of 0.5% (*w*/*v*)) was added to each well to incubation for 4 h at 37 °C in 5% CO_2_ environment. After removal of the medium, 150 μL of DMSO were mixed in each well. The cell viability was calculated as percentage relative to the control which was treated with culture medium only. The absorbance was measured by a microtiter plate reader at 570 nm.

To assure the definite cell growth inhibitory effects of total flavonoids extracted from germinated tartary buckwheat, different concentrations of BWTFs were added to cell culture medium (6.25, 12.5, 25, 50, 100 μg/mL, final concentration). MTT assay was used to evaluate the cell viability [35]. The concentrations required to inhibit growth by 50% (IC50 values) were calculated from cytotoxicity curves by SPSS [36].

### 4.7. UPLC-PDA Analysis Conditions

The components of the total flavonoids of tartary buckwheat were analyzed by Shimadzu UPLC-PDA system [37]. The components of BWFTs were determined by reference to quercetin and rutin. Experiments carried out were repeated three times. Chromatographic conditions were as follows. Column: Inertsil ODS-SP (4.6 × 250mm, 5 μm, Tokyo, Japan), column temperature: 30 °C, flow rate: 1 mL/min, solvent A: 0.1% formic acid in water, solvent B: acetonitrile. Injection volume: 5 μL, detection wavelength: 368 nm. Gradient elution procedure: 0–5 min, B: 50–55%; 5–8 min, B: 55–60; 8–20 min, B: 60%.

### 4.8. Apoptotic Assay

The state of the cells was detected by flow cytometry. The MGC80-3 cells were incubated in samples for 48 h and treated according to the requirement of the Apoptosis and Necrosis Assay Kit. Apoptosis and necrosis rate were determined by red and blue fluorescent channels. The positive control was 5-FU.

### 4.9. Protein Assay and Western Blotting

The cells after the intervention were collected [38] and the cell lysis buffer for Western and IP was added to extract the protein. The concentration was determined by the BCA Protein Assay Kit. Equal amounts of cell lysate from various treatments (100 mg of protein) were resolved by sodium dodecyl sulfate polyacrylamide gel electrophoresis [39], and then electrophoresis transferred onto polyvinylidene fluoride (PVDF) membranes. The PVDF membrane was placed in Blocking Buffer for 60 min at room temperature. Primary antibodies: β-actin and Bax (1:1000) were diluted in Primary Antibody Dilution Buffer; Bcl-2 and caspase-8 (1:500) were used as the primary antibody. The membrane was incubated with the primary antibody at 4 °C overnight. The second antibody, HRP-goat anti-rabbit IgG (H + L) (diluted 1: 1000 with Secondary Antibody Dilution Buffer), was added and shaken for 2 h at room temperature. Subsequently, the membranes were washed three times (10 min each time). The band molecular weight was analyzed by an X-ray photograph.

### 4.10. Statistical Analysis

The statistical analysis of the results was carried out by SPSS 17.0 statistical software. The results were expressed as the mean ± SD of at least three experiments. While comparing the change, the data was analyzed by one-way ANOVA, followed by the LSD to detect significant difference between different groups. The level of significance was set at *p* < 0.05.

## 5. Conclusions

The current work indicates that treatment with EF leads to an increase of total flavonoid contents. Human gastric carcinoma cell line (MGC80-3) was used to examine the anti-tumor properties of tartary buckwheat total flavonoids. The extracts (BWTF3) demonstrated the most obvious anti-tumor activity, comparatively. This seems to be due to the fact that the BWTFs reached a ratio in which the inhibitory activity was optimized after EF treatment. The biological value of tartary buckwheat can be further improved by electric field treatment, and it can be better applied to cancer prevention.

## Figures and Tables

**Figure 1 molecules-24-03092-f001:**
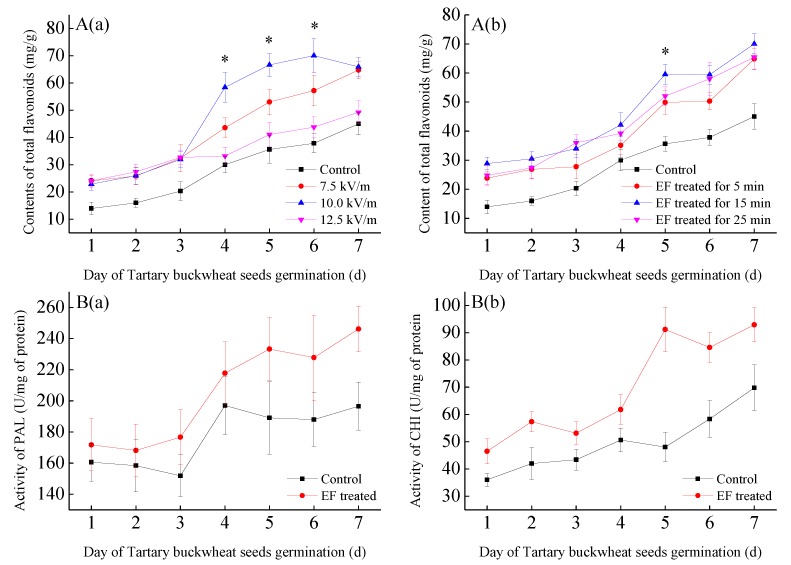
Total flavonoids in electric field (EF)-treated tartary buckwheat seeds (**A**). The effects of different intensities (7.5–12.5 kV/m) (**a**) and EF stimulation times (5–25 min) (**b**) were respectively measured and compared. One way analysis of variance (ANOVA) with Least-Significant Difference (LSD) was used to determine significant differences with respect to control and EF-treated groups. *, *p* < 0.05. The seeds untreated were control. (**B**) Enzyme activities of phenylalanine ammonia lyase (PAL) (**a**) and chalcone isomerase (CHI) (**b**). EF (10 kV/m, 15 min) significantly improved the content of total flavonoids (4–6 days). The untreated seeds were the control. Values were expressed as the mean ± standard deviation (SD) from experiments in triplicate.

**Figure 2 molecules-24-03092-f002:**
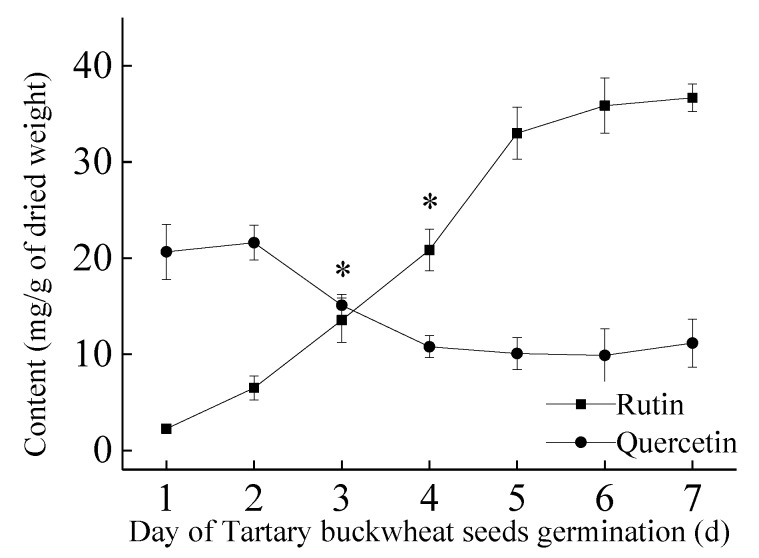
Changes in rutin and quercetin contents during tartary buckwheat germination. During the germination process, the content of rutin was significantly increased, and the content of quercetin was relatively stable. Values were expressed as the mean ± SD from triplicate experiment. One-way ANOVA with LSD was used to determine significant differences over time. *, *p* < 0.05.

**Figure 3 molecules-24-03092-f003:**
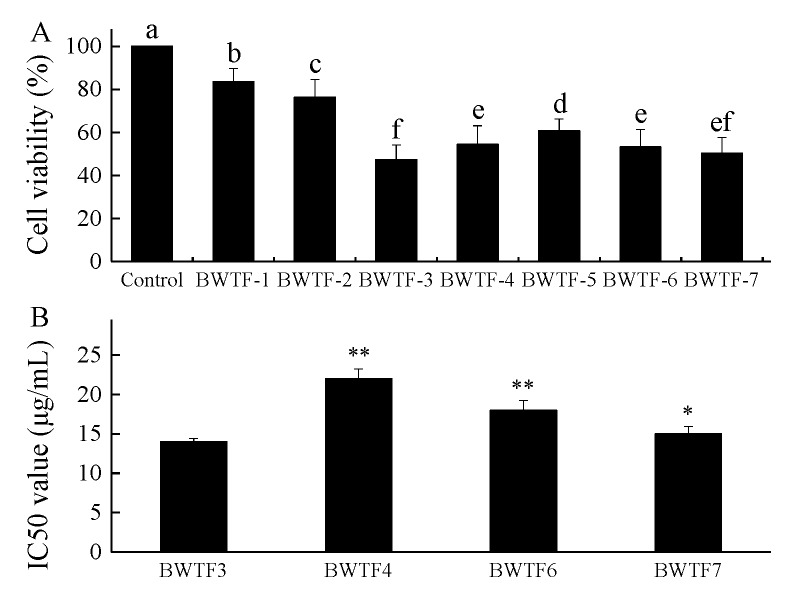
Anti-tumor activity of buckwheat flavonoids. (**A**) Total flavonoids of tartary buckwheat (BWTFs) were tested for inhibition activity on MGC 80-3 cells. BWTFs exhibited statistically inhibition activity. One-way ANOVA with LSD was applied for multicomparison. Bars with different letters indicate a statistical difference (*p* < 0.05), respectively. Bars bearing same letter mean not significant (*p* > 0.05). The control was treated with medium only. (**B**) The 50% inhibitory concentrations (IC50s) of BWTFs. One-way ANOVA with LSD was used to determine significant differences. * or ** indicates significant levels of differences at *p* < 0.05 or *p* < 0.01 with respect to BWTF3, respectively. Values were expressed as the mean ± SD from experiments in triplicate.

**Figure 4 molecules-24-03092-f004:**
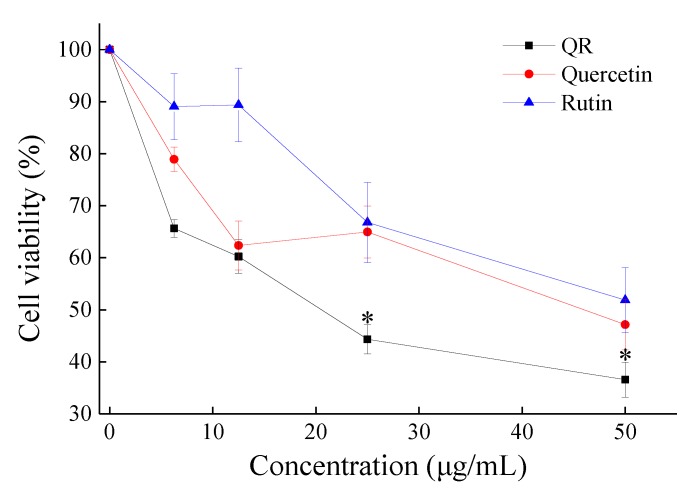
Effect of different concentration complexes on cell viability of MGC80-3. MGC80-3 cells were incubated with samples at indicated concentrations for 24 h, followed by determination of cell viability with MTT. QR (quercetin:rutin = 1:1). Values were expressed as the mean ± SD from experiments in triplicate. One-way ANOVA with LSD was used to determine significant differences within a given treatment. *, *p* < 0.05 vs. quercetin and rutin.

**Figure 5 molecules-24-03092-f005:**
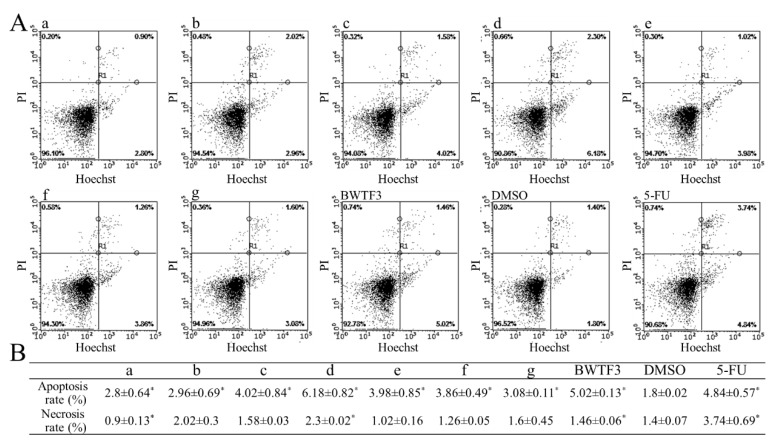
Effect of flavonoids on apoptosis of MGC80-3 cells. (**A**) the result of flow cytometry test. (**a**–**g**) indicates different proportion of quercetin and rutin (**a**, rutin:quercetin = 50:0; **b**, rutin:quercetin = 40:10; **c**, rutin:quercetin = 30:20; **d**, rutin:quercetin = 25:25; **e**, rutin:quercetin = 20:30; **f**, rutin:quercetin = 10:40; **g**, rutin:quercetin = 0:50 (μg/mL)). Dimethyl sulfoxide (DMSO), negative control; 5-fluorouracil (5-FU), positive control. (**B**) The result of apoptosis and necrosis in MGC80-3 cells. The definitions of **a**–**g** are as above. Values were expressed as the mean ± SD from experiments in triplicate. The data was analyzed by one-way ANOVA followed by the LSD post hoc test. * indicates significant levels of differences at *p* < 0.05 with respect to negative control (DMSO).

**Figure 6 molecules-24-03092-f006:**
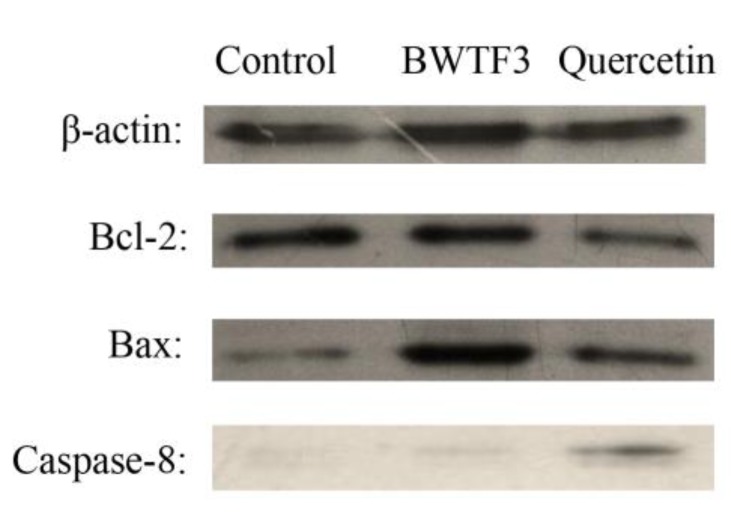
Effect of ethanol Extracts from tartary buckwheat sprouts powder on the expression of apoptosis-related protein in MGC80-3 cells.

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
