# Peer review of "The Effect of Tartary Buckwheat Flavonoids in Inhibiting the Proliferation of MGC80-3 Cells during Seed Germination"

_molecules, 2019, doi:10.3390/molecules24173092_

Round 1

Reviewer 1 Report

molecules-558475 review

Overall:

There are significant deficiencies in English grammar, spelling and usage throughout the manuscript, these need to be corrected. A professional manuscript editor should be employed to correct these issues. 

Abstract:

Fairly well represents the paper, apart from numerous issues with English usage and expression.

Introduction:

Reasonably clear apart from numerous issues with English usage and expression. Sets the study in context.

Methods: Generally sound but no description of the vehicle controls. What volume of the extract and vehicle was added to the 96 well plates? There is inadequate information on the Bliss method to determine IC50’s and there are no details in the cited reference.

Statistics:

Generally appropriate for comparisons within treatments, but no indication of the statistics used for comparisons between treatments.  Statistical significance is not referred to in results and only once in the figure legends.

Results:

Generally describe the findings reasonably but needs statistical support.  Eg. “The enhancements of total 85 flavonoids stimulated by 10 kV/m were superior” is not supported by any statistical analysis.

Page 5 Line 160: “This 160 may be due to the quercetin has a smaller molecular weight than rutin which penetrates into cell 161 membranes more easily” this is almost certainly not the case, it is most likely they are acting on extracellular receptors and that the rutin glycoside is interfering with binding.

Figures:

Figure 1. No statistics are reported in the figure legend.  Statistics need to be indicated to support the reported results. (eg. Values reported as mean ± SD, N= x. * P <0.05 from day 1 control one way ANOVA with LSD post hoc test). Although this is stated in the statistics section each figure should be stand alone. Statistics for comparisons between treatments need to be shown as well.

Figure 2. Requires statistics as above.

Figure 3. What are the significance stars in relation to? You cannot do significance from control as this is normalised data can the control has an artificial zero variance. Is BWFT-3 significantly different from the other fractions?

Table 1. No error ranges on the IC50 values

Figure 4. Y axis title is incorrect; it should be Cell viability % control.  Again, no statistics are presented. Curves should be fitted not done as dot to dot connections.

Figure 5, letters a-g need to be defined in the figure legend. Figure legends need to stand alone.

Discussion: Rather short and will need to be revised once appropriate statistics are performed. Again, numerous issues with English usage and expression. Potential benefits to gastric cancer patients are speculative, should be more nuanced  and should come after the discussion on effects on cancer cells.

Reviewer 2 Report

The authors in this manuscript evaluated the effect of electric field on tartary buckwheat flavonoids production during seed germination and their effects on inhibiting proliferation of human gastric cancer cells. Appropriate methods are being used. Studies are conducted systematically. Results support the rationale and aims of the study. However, there are number of concerns that needs attention.

1.       Why do authors assume that only positive effects will be seen due to electric field treatment of seeds?

2.       Language needs significant improvement.

3.       How many different samples of buckwheat seeds were evaluated? Each batch may possess different levels of flavonoids.

4.       In the figures, how many samples/group were analyzed?

5.       Authors optimized EF intensities and times separately. However, authors should consider statistical optimizations that will show optimal intensities and times at the same time.

6.       Control is missing in Fig 2.

7.       In figure 3, Why is there a difference in the anti-tumor activities of different samples of BWTFs.

8.       In figure 5, What is the concentration of quercetin and rutin in BWTF?

9.       In figures 4, 5 and 6, controls i.e non-EF treated sample extracts are missing.

10.   Figure 6, quantification of the western blots shall be done.

Reviewer 3 Report

REVIEW  molecules-558475

The effect of the Tartary buckwheat flavonoids in inhibiting proliferation of MGC80-3 cells during the seeds germination

Major questions:

The experiment is very interesting, but a lot of mistakes crept into the manuscript mainly in the methodical part. First of all, is the fraction called ‘total flavonoids’ is  really composed of just two flavonoids (?) or the mentioned extract / fraction has an unidentified composition? Was chromatographic analysis carried out to assess if the ethanol-water extract contains just flavonoids and could be synonymously named flavonoids? Did the authors determine what flavonoids were present in this extract? Not only these two studied – quercetin and rutin? Or maybe the two were dominant? Please explain why the MTT test was chosen for the cell viability?

Minor corrections:

- Line 2 is ‘tartary’ – should be ‘Tartary’ (capital letter)

- Line 251 – ‘… 5-fluorouracil were purchased from China…’ what is the name of the company (producer/distributor)?

- Lines 251-258 – some of the chemicals and reagents are written beginning with the capital letter, some of them with the small letter

- Line 260 – authors should provide the geographical position of the cultivation plot of the plants, which were the source of the seeds

- Line number 263 – what is the correct subtitle?

- Line 264 – what are the parameters of the extraction process?

- Line 266 – what kind of filtration?

- Lines 267-274 – this paragraph should be described with more details, is unreadable

- Lines 291-297 – this paragraph also should be written with more details about the equipment and analysis conditions. There is no information about the quality of the chemicals and solvents applied, the standards used for identification of sample components. The manuscript does not indicate what the operating parameters of the detection systems used for HPLC (parameters of the ionization source, diode array, etc.). It is not clear how the quantitative analysis of metabolites was carried out. What were the parameters of the calibration curves and the reproducibility of the method, how many analytical / biological repeats?

- figure 3 – please explain the term ‘control’ and below the figure put the explanation of the mentioned abbreviations

- table 1 – check the line number 152 and 153, the description of the table is also unreadable (too many abbreviations), the samples of the first column should be described below the table

- Lines 9 and 26 – dot after Gaertn?

- Line 9 – space between Fagopyrum and tataricum

2.3. Quantitative Analysis – mg/g of dried weight?

- figure 5 – the symbols (from a to g) should be explained below the table

Generally, the paper contains several minor typos, language errors and ambiguous errors.

The reference list should be prepared according to the instruction for authors. Many mistakes.

Round 2

Reviewer 1 Report

Overall:

The authors are to be commended for the effort they have made revising this manuscript. However several deficiencies remain. There are still significant deficiencies in English grammar, spelling and usage throughout the manuscript, these need to be corrected. A professional manuscript editor should be employed to correct these issues. 

Methods: The vehicle controls are still inadequately described. What was the vehicle control for the 96 well plates? The cited reference on the Bliss method to determine IC50’s has no detail and refers to another paper which also has no details which in turn refers to a history of William Ostler which is not about IC50’s at all.

Statistics:

Generally appropriate for comparisons within treatments, but no indication of the statistics used for comparisons between treatments (I am assuming then LSD test is for comparisons within a given treatment, as there is no indication of the use of two way ANOVA which should be used for between treatment effects).  Statistical significance is now referred to in results but in the figure legends, it is still unclear (see figures comments).

Figures:

Figure 1. Statistics are now reported in the figure legends but the statistical tests are still missing.  There are no significance stars on the points to determine which points are significant from which. Each figure should be stand-alone. Statistics for comparisons between treatments also need to be shown (i.e one way ANOVA with LSD was used to determine significant differences over time and two way ANOVA was used to determine significant differences between control and EF treatment).

Figure 2. Requires statistics as above (ie significance starts and a statement like one way ANOVA with LSD was used to determine significant differences over time).

Figure 3. What do the letters mean (Bars with different letters indicated statistical difference)? In panel A is a significantly different from b? c? What does ef mean? Do the letters in panel B mean the same thing as panel A? Each figure should be standalone and should state something like one way ANOVA with LSD was used to determine significant differences between fractions and the letter meaning shown explicitly.

Figure 4. This figure would be better with fitted curves rather than bars. Each figure should be standalone and should state something like one way ANOVA with LSD was used to determine significant differences within a given treatment. The letters on the columns need to be defined (eg a= P<0.05 vs Quercetin C= P < 0.05 vs Quercetin and Rutin).

Figure 5, letters a-g are now defined but the superscript letters a-d need to be defined in the figure legend. Figure legends need to stand alone (see previous comments on statistics).

Discussion: Much improved but there are still numerous issues with English usage and expression. Potential benefits to gastric cancer patients remain speculative.  For example, this statement “Flavonoids in plants are effective anticancer substances, such as Babu et al. [27] demonstrated that flavonoid from the Syzygium alternifolium significantly inhibited the cell proliferation of human gastric cancer cells.” This is still just in tissue culture, there is no in vivo demonstration of effectiveness in animal models, let alone human cancer and the Babu et al paper is about using these compounds as a scaffold for making anticancer compounds, rather than being used as anticancer compounds themselves. This part of the discussion needs to be more careful.

Reviewer 2 Report

I do not have any more comments.

Author Response

Thank you very much.